# Naupliar exposure to acute warming does not affect ontogenetic patterns in respiration, body size, or development time in the cosmopolitan copepod *Acartia tonsa*

**Mathew Holmes-Hackerd**[1], **Matthew Sasaki**[1,2]*, **Hans G. Dam**[1]

1 Department of Marine Sciences, University of Connecticut, Groton, Connecticut, United States of America,
2 Department of Biology, University of Vermont, Burlington, Vermont, United States of America

\* matthew.sasaki@uconn.edu

**Data Availability Statement:** All data and code relevant to this study are available without restriction from a Zenodo repository (https://doi.

## Abstract

Short-term, acute warming events are increasing in frequency across the world's oceans. For short-lived species like most copepods, these extreme events can occur over both within- and between-generational time scales. Yet, it is unclear whether exposure to acute warming during early life stages of copepods can cause lingering effects on metabolism through development, even after the event has ended. These lingering effects would reduce the amount of energy devoted to growth and affect copepod population dynamics. We exposed nauplii of an ecologically important coastal species, *Acartia tonsa*, to a 24-hour warming event (control: 18°C; treatment: 28°C), and then tracked individual respiration rate, body length, and stage duration through development. As expected, we observed a decrease in mass-specific respiration rates as individuals developed. However, exposure to acute warming had no effect on the ontogenetic patterns in per-capita or mass-specific respiration rates, body length, or development time. The lack of these carryover effects through ontogeny suggests within-generational resilience to acute warming in this copepod species.

## Introduction

Copepods are globally distributed and ecologically important in aquatic ecosystems. As primary consumers, copepods represent a crucial link between primary producers and higher trophic levels. As the primary food source for many fish species, copepods have a direct effect on fisheries recruitment [1,2]. Furthermore, copepods are arguably the most abundant pelagic animals [3] and dominate zooplanktonic biomass. Through their production of carbon and nutrient-rich fecal pellets, and their diel vertical migration, copepods accelerate vertical transport of particulate and dissolved matter into deeper waters [4,5]. Altogether, copepods play a vital role in global carbon and nutrient cycling. A robust understanding of how copepods are affected by increasing temperature is a crucial component of predictions about their various roles under future climate scenarios.

org/10.5281/zenodo.7434983). In addition to this repository, data and code are actively maintained on GitHub (https://github.com/ZoopEcoEvo/resp_through_dev/tree/v1).

**Funding:** This research was supported by a National Science Foundation Grant (OCE 1947965) awarded to HGD. The funders had no role in study design, data collection and analysis, decision to publish, or preparation of the manuscript.

**Competing interests:** The authors have declared that no competing interests exist.

Organismal responses to climate stressors can be broadly separated into two categories: across-generation (evolutionary change) and within-generation (individual variation and phenotypic plasticity). These timescales are intertwined, however, with the latter providing the raw material for evolutionary change [6,7]. Fine-scale observations of within-generation responses to shifts in environmental conditions can, therefore, provide important insights into both the vulnerability and resilience of populations to climate change. Anthropogenic climate change is predicted to cause a rise in ocean temperatures of ~1.5˚C over the next 20 years [8], which may have a variety of adverse effects on marine taxa. Perhaps most impacted by this warming will be small ectotherms [9,10]. In addition to rising average temperatures, short-term heat events (e.g. marine heat waves) are increasing in intensity, duration, and frequency [11,12]. Increases in temperature over these shorter timescales pose challenges to organisms that are unique from those presented by the long-term increase in average temperature; namely, the increase in temperature is often more extreme and may occur over within-generation time scales, even in short-lived marine organisms. Given the potentially short timescale of these events, a population's initial response to acute warming is likely to be driven by phenotypic plasticity. Importantly, there may also be variation in individual responses, due to standing genetic diversity within populations. This individual-level variation may shape the impact of increased temperatures on population persistence over short timescales. Outside of low latitudes, most copepods appear to maintain large buffers between environmental and lethal temperatures [13]. As such, understanding how acute sub-lethal exposure to warming affects copepod performance traits is necessary to help predict population dynamics as these events become more frequent. It is also important to separate the effects of acute events from chronic exposure, especially in taxa like copepods where the duration of exposure may extend across developmental stages.

Temperature is known to have strong effects on copepod life history traits [14–16]. For example, temperature alone was found to explain >90% of the variance in growth rates of copepods collected from environments ranging from -1.7 to 30.7˚C [15]. Similar dependence on temperature is found in other copepod life history traits such as egg production [17] and ingestion rates [18,19]. Copepod metabolism is also highly temperature sensitive [16]. Broadly, metabolism is understood as the sum of all chemical reactions that occur within a living organism because it entails the rate of energy uptake, transformation, and allocation [20]. Metabolism is a key bioenergetics function that provides insight into the control of individual growth [21,22].

Aerobic respiration rates are often used as a proxy for metabolic rates because almost all oxygen consumption is used to support energy production. By focusing on the relationship between temperature and copepod respiration we may therefore obtain an understanding of temperature effects on copepod metabolism and performance at the whole organism level. Copepod metabolic rates tend to increase with increasing temperature [16,23,24]. At high temperatures, additional increases in metabolic rates may stem from the energetic costs associated with mounting the heat shock response [25]. For instance, effects of exposure to elevated temperatures can be buffered by an increase in intracellular concentrations of heat-shock proteins [26]. This may persist, with carryover effects on copepod metabolism, even after the exposure has ended the upregulation of heat shock protein gene expression in adult *Acartia tonsa*, for example, can persist for up to 30 hours past the warming event [27]. The metabolic implications of this response strategy remain unexplored.

Respiration rate is also often related to body size. Per capita respiration rate, R, scales to body mass, M, as: $R = aM^{3/4}$ [28]. Hence, the weight-specific respiration rate, R/M, decreases with increasing body mass. This pattern holds for copepods [29,30], with some caveats: the pattern accurately describes copepodite, but not naupliar, stages [31]. Despite large relative

changes in mass through the six naupliar stages, mass-specific respiration remains relatively constant [31]. This deviates from the expectation of decreasing mass-specific respiration as the nauplii grow. Furthermore, a large increase in metabolism between the final naupliar stages and first copepodite stage has been observed [31], which is again counter to the expectation that mass-specific respiration rate should decrease with increased mass. The stark metabolic differences between copepod life stages suggest there may also be differences in the response to elevated temperature. Generally, early life stages of marine organisms are less tolerant to elevated temperature and thus may act as demographic 'bottlenecks' for populations in response to acute warming events (although this is not universal, and copepod nauplii may be more resistant; [32]). Quantifying the effects of exposure to warming during the naupliar stages on ontogenetic changes in respiration in individual copepods may provide insight into how copepod populations may respond to the increasing threat of short-term sublethal warming events.

The goals of this study are to test the resilience of copepods to acute warming experienced during the naupliar stages, and to quantify potential carryover effects of short-term sublethal warming. We used high sensitivity respirometry to measure individual respiration rates of the calanoid copepod, *A. tonsa*, throughout development after an initial naupliar warming treatment; thus, providing a fine-scale perspective on the effects of non–lethal warming exposure on copepod metabolism through development. The temperature treatment used was an increase of +10°C relative to rearing temperature for 24 hours. This timescale was much shorter than the conventional definition of marine heat waves because we were interested specifically in the effects of acute warming; a longer timescale exposure would introduce potential confounding effects of temperature on growth rates (as naupliar stages progress near daily). We hypothesized that naupliar warming treatment would lead to (i) higher energetic requirements and thus higher respiration rates during the subsequent copepodite stages, (ii) delayed development, with warming exposed individuals taking longer to progress through copepodite stages, and (iii) reduced body length across copepodite stages following the naupliar warming treatment.

## Methods

### Experimental organisms

*Acartia tonsa* nauplii were obtained from a laboratory culture initially collected in July 2021 at Esker Point, Groton, CT, USA (41.3209°N, 71.9996°W). The culture was maintained in an 11-liter container at 18°C in a temperature controlled environmental chamber under a 12:12 light:dark cycle. A small aquarium pump was used to ensure constant aeration of the culture. The culture was regularly fed ad-libitum a mixture of *Tetraselmis* sp., *Rhodomonas* sp., and *Thalassiosira weissflogii*, as used in previous studies [33]. Phytoplankton cultures were grown in F/2 medium and kept in the exponential growth phase under the same light and temperature conditions as the copepods.

### Temperature selection and naupliar respiration rates

Nauplii were exposed to a 28°C warming treatment at the initiation of the experiments. Preliminary experiments showed that this temperature did not exceed naupliar thermal tolerance while still inducing a strong metabolic response, indicating an impactful but sub-lethal heat challenge (Fig 1B). Selection of a non-lethal temperature was necessary in order to avoid confounding effects of selection for heat tolerant individuals on measurements later in development. The naupliar survivorship curve was generated following methods previously used for adult *A. tonsa* [33], with individual nauplii survivorship measured after a 24-hour exposure to a range of temperatures. A logistic regression was fit to this individual survivorship data to

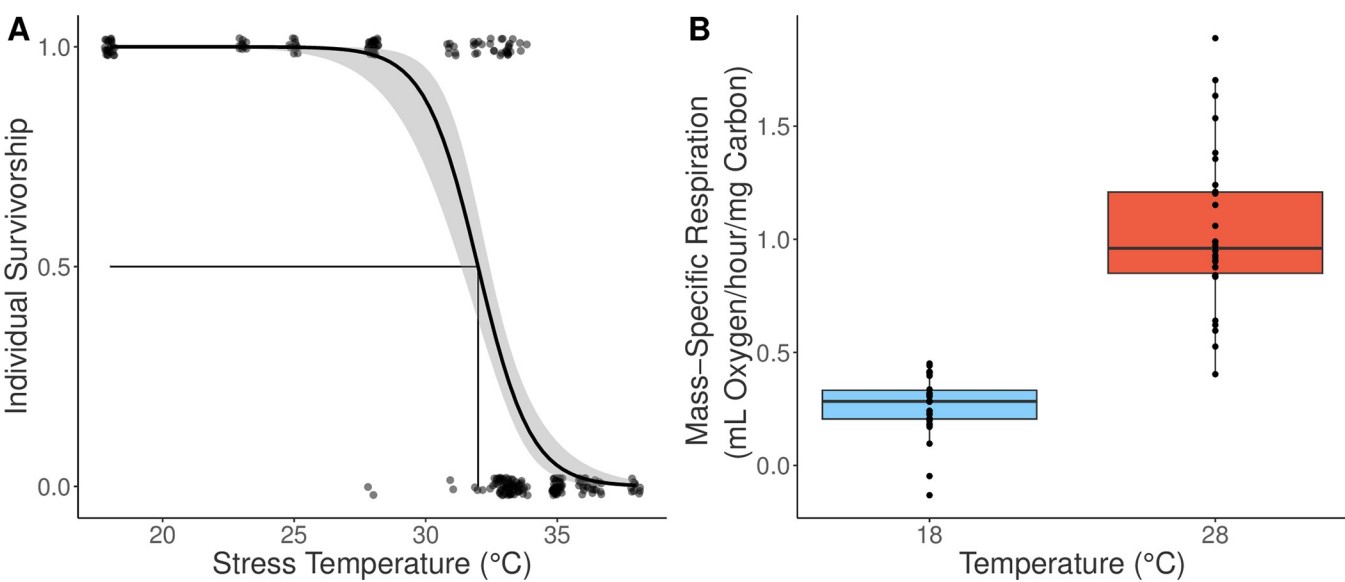

**Fig 1. Nauplii thermal survivorship and mass-specific respiration rates.** (A) Survivorship curve used to estimate naupliar thermal tolerance. Individual survivorship was determined after 24 hours exposed to a range of temperatures. A logistic regression was fit to this data to estimate the survivorship curve and LD50, the temperature inducing 50% mortality. (B) Box plot of mass-specific respiration rates (mL O2/hour/mg carbon) of nauplii at 18°C and 28°C. Sample size for both temperatures are 30 measurements per temperature. Measurements occurred for 8 hours, nauplii were reared in 18°C and not acclimated to 28°C ahead of time. The box plot reports median value (black line), interquartile range (colored range). Component respiration rate measurements are shown as individual points.

generate the thermal survivorship curve, which was used to estimate thermal tolerance (LD50, the temperature inducing 50% mortality). Respiration rate was estimated as the rate of oxygen depletion over time (see Respiration analysis below for further details). Oxygen measurements were taken using a Presens® oxygen sensor system, which previous studies have used to measure respiration rates of copepod nauplii [34]. In brief, the system uses airtight 2.8 mL vials in a dish atop an optical reader. Each vial contains a sensor spot made of oxygen-sensitive foil with an immobilized fluorescent dye along the base. These optical sensors measure oxygen concentration within the vial at discrete time intervals. Here we used 3-minute intervals to minimize the slight temperature fluctuations caused by the optical measurements. Naupliar respiration rates were measured for 8 hours in a standing incubator with no light. Three replicate 8-hour trails were conducted at both 18°C and 28°C. Each replicate included 10 vials containing 15 nauplii each and 2 control vials with no nauplii.

## Experimental design

Three replicate experiments were performed to test the effect of exposure to acute warming in the naupliar stages through subsequent developmental stages. To initiate each replicate ~200 newly hatched nauplii were isolated from the laboratory culture and evenly split into six 50 mL centrifuge tubes filled with 30 mL of replete food solution (800 μg C/L; 400 μg C/L each of Tetraselmis sp. and Rhodomonas sp.) These six tubes were randomly assigned to either control (18°C) or heat treatment (28°C) groups (n = 3 tubes per heat treatment per experiment). Each tube was loosely capped and placed into a dry heat bath (ThermoScientific Digital Drybath, Cat# 88870003) and maintained at the assigned temperature treatment for 24-hours.

Following the 24-hour exposure period, 30–40 nauplii were randomly selected from across the tubes (n = 15–20 per treatment) and placed in individual 20 mL petri dishes with replete food solution to monitor development. Petri dishes had been previously cured by soaking in

filtered sea water for 7 days with daily seawater replenishment. A subset of these dishes (n = 5–10 each treatment) was randomly selected and assigned to a handling control treatment. Handling controls were monitored daily (see Daily monitoring and trait measurements) but trait measurements were not made until the C6 stage, allowing us to examine potential handling effects on experimental copepods, which were measured and handled more regularly (see Daily monitoring and trait measurements). These experiments were blinded, with the treatment assignment process randomized and petri dish labels free of any indication of which treatment (control or warming treatment) the individual was exposed to.

## Daily monitoring and trait measurements

Each petri dish was checked using a dissection microscope every 24 hours for copepod survival. Light intensity was kept at the lowest possible level to visualize the copepod while minimizing potential light stress. Each surviving individual was carefully transferred into another cured and labeled 20 mL petri dish containing 10 mL fresh replete food solution. Its previous dish and solution were checked for shed molts to determine if or when the individual had progressed to the next development stage and to estimate stage duration (days).

Each individual's respiration was measured once per developmental stage immediately following a molt being found (see Copepodite respiration rates section). Following respirometry, the copepodite was photographed using an inverted microscope (again under minimum light exposure) and transferred back to its individual petri dish. Body lengths were estimated from the photographs using ImageJ (http://imagej.org/) and then converted into carbon mass (ng C) using an empirical relationship between mass and length [35].

## Copepodite respiration rates

Respiration rates of the six copepodite stages (C1-C6) were measured using the same Presens℞ system used to assess naupliar respiration. For copepodite measurements, single individuals were placed in each vial with the same ad libitum food solution (800 μg C/L evenly split between Tetraselmis sp. and Rhodomonas sp.) to prevent starvation effects on respiration rates [35,36]. During respiration measurements, oxygen concentrations were measured every three minutes for 8 hours. All measurements were made at 18˚C in the dark using a standing incubator. This measurement period was sufficient to detect oxygen drawdown, while minimizing the amount of experimental handling each individual experienced. Special care was taken to remove all air bubbles from the vials as their presence interferes with the sensor's capability to accurately measure oxygen concentration.

## Respiration analysis

Respiration rates were calculated from the time-dependent oxygen concentrations in the vials using the R statistical and computing software environment (version 4.1.3; [37]) and the respR package (version 2.0.2; [38]). Respiration rates were calculated against two or three control vials filled with the replete food solution but without a copepod, to account for background changes in oxygen concentration. Respiration rate was then calculated as the slope of oxygen concentration against time, derived from a linear regression. Oxygen depletion within the control vials was negligible in comparison to the depletion within vials containing a copepod, and thus any potential changes in phytoplankton cell abundance due to grazing during the incubation period were not considered (S1 Fig in S1 File). The first 350 minutes of oxygen measurements were excluded from the regression analyses because of curvature in the oxygen profiles, likely produced while the oxygen sensor spot became saturated or during temperature acclimation between the laboratory space and standing incubator. Excluding this initial time period

ensured linearity of the oxygen drawdown, confirmed by visual inspection of the oxygen traces, which is necessary because calculation of respiration rates assumes linear drawdown. The estimated body mass of each copepodite (ng carbon) was used to calculate mass-specific respiration rates (mL oxygen/ hour/ mg carbon).

### Hypothesis testing

To test for differences in survival through development between treatment and control groups a Kaplan-Meier survival analysis was conducted. To further explore potential effects of treatment and experimental replicate on survival a Cox proportional hazard model was applied to the data. Separate linear mixed effects models were used to test the effect of the naupliar warming treatment on copepodite traits (size, respiration rate, and total development time) and whether treatment effects depended on sex and life stage. These models included experimental replicate (n = 3) and, within each replicate, the copepod individual as random effects. For copepodite development time (days from C1 to C6), a linear model was used with fixed effects of sex and naupliar warming treatment and random effect of experimental replicate. To test for handling artifacts on the three variables, the respiration rates and body size of handling controls at C6 were compared to the C6 measurements of the serially measured individuals. Copepodite development time was also compared between actively handled and handling control individuals. Post-hoc comparisons of estimated marginal means from mixed effect models were used to test specific differences between groups and explore significant interactions ('emmeans' R package, v 1.7.5). Model assumptions were validated by visually inspecting residual and normal-quantile plots. Several models included a weighted variance structure to account for unequal variances among treatment groups [39] using the R package 'nlme' (version 3.1–160).

### Results

Nauplii had a thermal tolerance value (measured as LD50) of 32˚C (Fig 1A). Naupliar respiration rates increased when exposed to 28˚C (treatment) compared to 18˚C (control) (treatment: 1.04±0.06, control: 0.29±0.002 mL O2/hour/individual (mean±S.E.); Fig 1C). This difference in respiration rates yielded a Q10 of 3.6, which is greater than typically published estimates of copepod respiration (Q10 = 1.8–2.1; [16]) consistent with the idea that the warming temperature chosen induced a strong metabolic response but was not lethal to the nauplii.

Measurements were made for a total of 80 individuals as they progressed through development. Of these, 68 successfully reached maturity (stage C6) and could be reliably sexed (50 females and 18 males). Body length increased through development in both treatment and control copepodites (Fig 2). Individual females grew from 0.33±0.002 mm in C1 to 0.85±0.007 mm in C6, and males grew from 0.35±0.005 mm in C1 to 0.76±0.006 mm in C6 (mean±S.E.). Per capita respiration also increased throughout development (Fig 3A), ranging by a factor of four from C1 to C6. As expected, body length (and thus calculated mass) increased at a higher rate than respiration as copepods developed, resulting in a decrease in mass-specific respiration throughout development (Fig 3B). Survival in both groups was high (S2 Fig in S1 File), with 80% (control) and 63% (treatment) of individuals reaching maturity, with no significant effect of treatment on survival (Kaplan-Meier test p = 0.21, Cox hazard ratio p = 0.228; S2 and S3 Figs in S1 File). Survival was the same in all three experimental replicates (replicate 1 vs. 2 p = 0.283, replicate 1 vs. 3 p = 0.407, S3 Fig in S1 File).

In contrast to the strong effects of copepodite stage on measured traits, we found no indication of an effect of naupliar warming treatment on mass-specific respiration rates (p = 0.7166, Table 1), body size (p = 0.1316, Table 2), or development time (p = 0.1943, Table 3). Our first

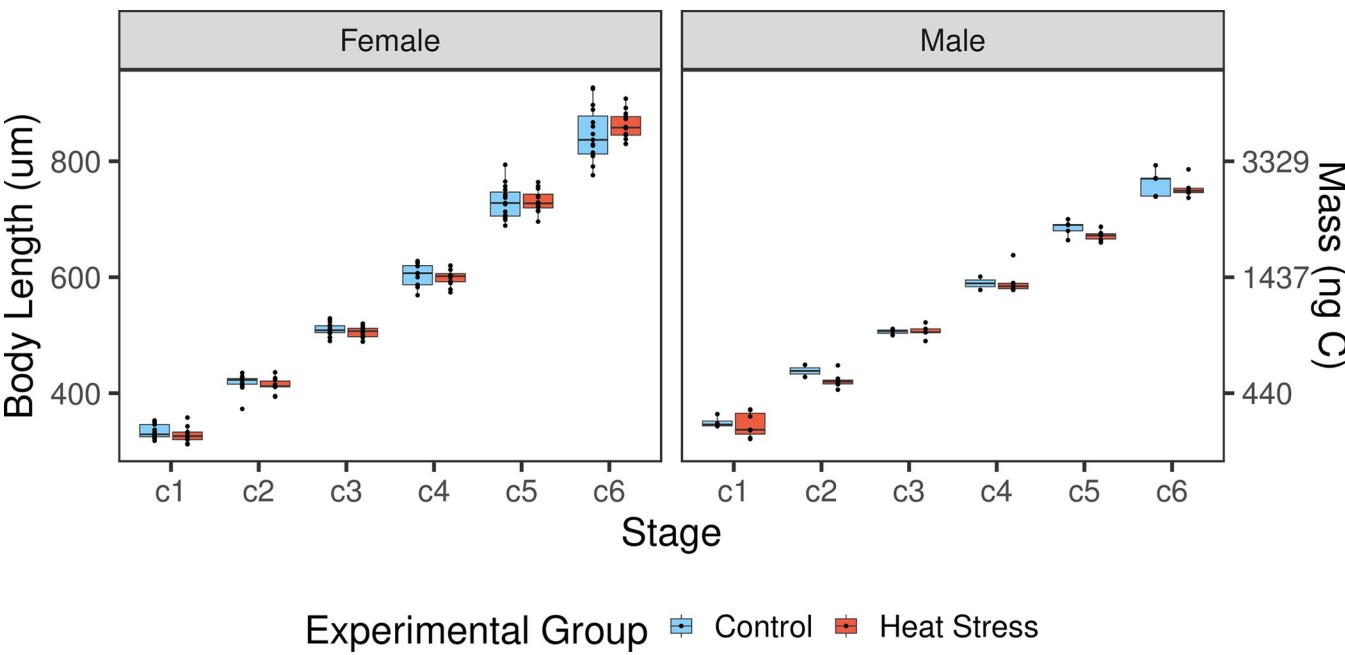

**Fig 2. Stage specific body lengths of male and female *Acartia tonsa* copepodites.** Body lengths (mm) for individuals from each experimental group at the various copepodite stages. Different treatments are indicated by fill color. Individual body size measurements are shown as points. Box plot components as in Fig 1B.

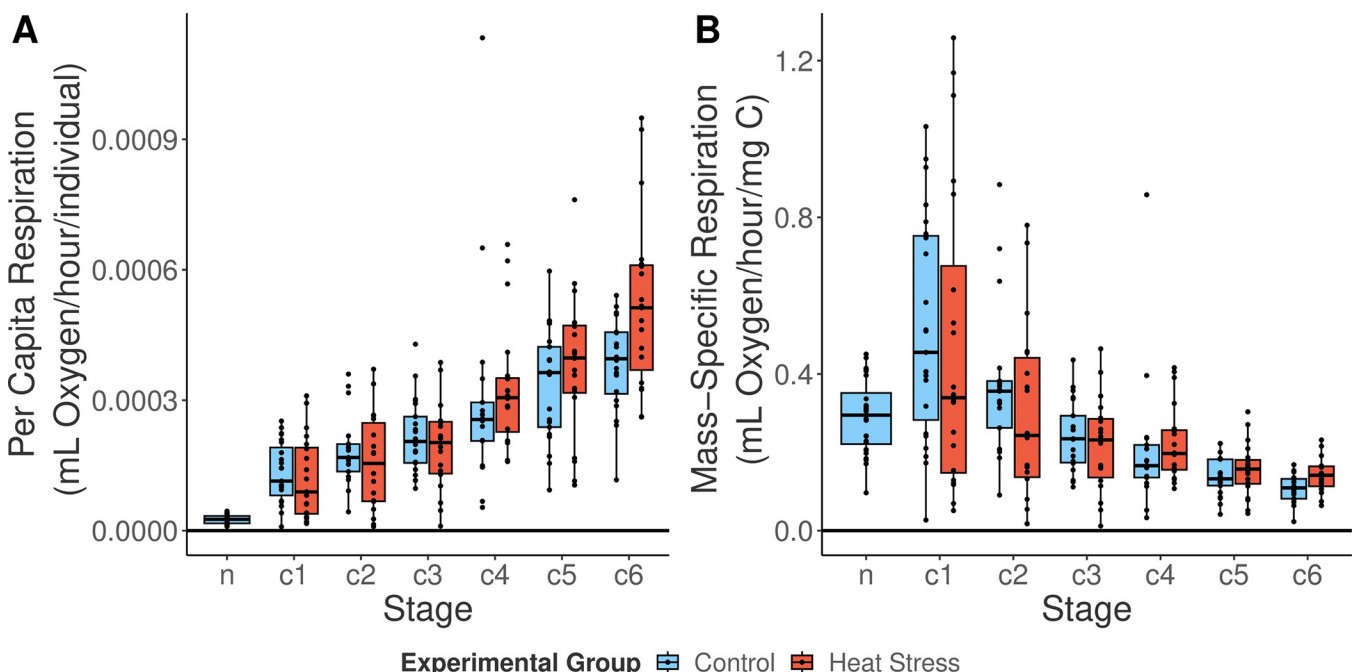

**Fig 3. Per capita and mass-specific respiration rates of *Acartia tonsa*.** (A) Per capita respiration rates (mL O2/hour/ind.) of pooled nauplii (n) and individual copepodite stages (C1-C6). (B) Mass-specific respiration rates (mL O2/hour/mg carbon) of pooled nauplii and copepodite stages. Different treatments are indicated by different fill colors. Individual respiration rate measurements are shown as individual points. Boxplot components as describes in Fig 1B.

**Table 1. ANOVA output of mass-specific linear mixed effects model.**

| Effect | DFn, DFd | F | P |
|---|---|---|---|
| Stage | 5, 145 | 30.79 | **<0.0001** |
| Sex | 1, 35 | 1.844 | 0.1048 |
| Treatment | 1, 35 | 4.537 | 0.7166 |
| Stage:Sex | 5, 145 | 2.261 | 0.3401 |
| Stage:Treatment | 5, 145 | 0.5515 | 0.9959 |
| Sex:Treatment | 1, 35 | 0.4737 | 0.3183 |
| Stage:Sex:Treatment | 5, 145 | 1.014 | 0.4076 |

Summary of results from a linear mixed effects model testing the effects of naupliar warming exposure on mass-specific respiration rates. Warming treatment, sex, and copepodite stage and their interactions were included in the model as fixed effects, while experiment replicate and individual copepod were included in the model as random effects.

hypothesis that naupliar exposure to warming would cause increased mass-specific respiration rates was tested using a linear mixed effect model, which indicated that mass-specific respiration rates varied between development stages (p<0.0001), but not between sexes or experimental group (Table 1). Our second hypothesis that, relative to control individuals, body lengths would decrease after naupliar warming exposure was also not supported. The linear mixed effect model for this hypothesis showed that body length varied between stages (p<0.0001) and sexes (p<0.0001), but not between treatment groups (Table 2). Post hoc tests show males were larger than females in C1 (p = 0.0120, S1 Table in S1 File), both sexes were the same size in C2-C4, and females were larger in C5 and C6 (both p < 0.001, S1 Table in S1 File). Finally, the third hypothesis, that total development time would increase in response to naupliar warming treatment, was also not supported (Fig 4, Table 3); however, development time was sex-dependent, with males developing faster (p<0.0001, Table 3).

Handling controls were used to explore potential effects of re-occurring experimental handling on respiration and body length. All comparisons between handling groups were for C6 individuals, as neither respiration rates nor body lengths were measured in the intermediate copepodite stages in the handling controls. Three-way ANOVAs indicated no effect of handling on mass-specific respiration rates (p = 0.295, S2 Table in S1 File), body lengths (p = 0.276, S3 Table in S1 File), and copepodite development time (p = 0.092, S4 Table in S1 File).

**Table 2. ANOVA output of stage-specific body lengths linear mixed effects model.**

| Effect | DFn, DFd | F | P |
|---|---|---|---|
| Stage | 5, 145 | 4103 | **<0.0001** |
| Sex | 1, 35 | 30.08 | **<0.0001** |
| Treatment | 1, 35 | 2.424 | 0.1316 |
| Stage:Sex | 5, 145 | 49.52 | **<0.0001** |
| Stage:Treatment | 5, 145 | 0.4730 | 0.6094 |
| Sex:Treatment | 1, 35 | 0.7090 | 0.1476 |
| Stage:Sex:Treatment | 5, 145 | 2.868 | 0.0976 |

Summary of results from a linear mixed effects model testing effects of naupliar warming exposure on body lengths. Warming treatment, sex, and copepodite stage and their interactions were included in the model as fixed effects, while experimental replicate and individual copepods were included in the model as random effects.

**Table 3. ANOVA output of copepodite development time linear mixed effects model.**

| Effect | DFn, DFd | F | P |
|---|---|---|---|
| Sex | 1, 35 | 35.38 | **<0.0001** |
| Treatment | 1, 35 | 3.082 | 0.1943 |
| Sex:Treatment | 1, 35 | 1.195 | 0.2743 |

Summary of results from a linear mixed effects model testing effects of naupliar warming exposure on copepodite development time. Warming treatment and sex and their interactions were included in the model as fixed effects, while experimental replicate and individual copepod were included in the model as random effects.

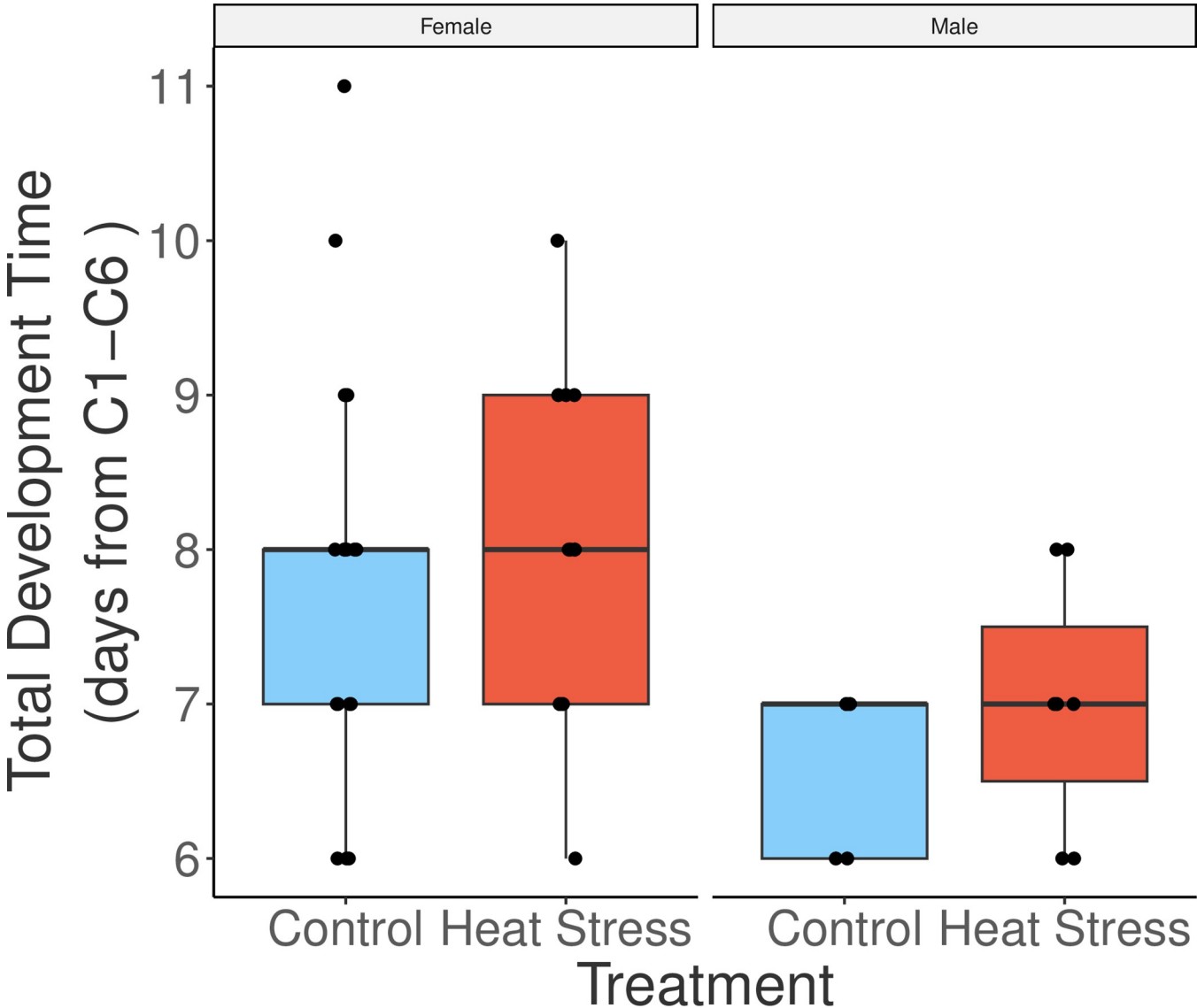

**Fig 4. Copepodite development time of male and female *Acartia tonsa*.** Boxplot of total copepodite development time from C1-C6 between experimental groups for both sexes. Box plot components as in Fig 1.

In summary, copepod respiration increased through development as did body length. Respiration rates increased at a slower rate than mass leading to decreasing mass-specific respiration rates through development. Females progressed through copepodite stages more slowly than males but ultimately reached a larger size. Neither the naupliar warming exposure nor experimental handling had an effect on mass-specific respiration rates, body size, or copepodite development time.

## Discussion

Exposure to sublethal warming can have various effects on copepods that linger past the stress event itself. Carryover effects are important to understand because they represent an increased energetic reallocation toward mediation of the stress and away from growth than estimated by just monitoring organismal performance during a stress event. By measuring and tracking individual copepods through their development we were able to explore potential effects of acute naupliar exposure to warming on ontogenetic patterns in metabolism and body size. In general, these ontogenetic patterns conformed to expectations, with the relative increase in respiration being less than that of body length, leading to an inverse relationship between mass-specific respiration and copepodite stage. Differences between sexes were observed for body size and development time, but not for mass-specific respiration rates. Interestingly, the naupliar warming exposure used here (+10˚C for 24 hours) did not affect subsequent respiration rate, body size, or development time, suggesting that *A. tonsa* development is resilient to intense but sub-lethal warming exposure. This suggests that effects of extended or repeated exposure may be of greater concern.

With no observable difference in respiration through ontogeny between the treatment and the control, our results suggest that there was no effect of acute warming in the naupliar stage on subsequent energy demand. We expected that the observed hardening tactic of upregulating heat shock proteins under increased temperature exposure in *A. tonsa* [40] would manifest as a shift in metabolic processes that would be sustained throughout development. We also hypothesized that production of heat shock proteins would divert energy from growth (decreasing body size) or development (increasing stage durations). While we did not measure heat shock protein production, past works show the high capacity for production and upregulation of heat shock proteins in response to temperature exposure in *A. tonsa* [27].

The lack of carryover effects of warming exposure on copepodites was surprising given the strong metabolic response of nauplii to the selected high temperature treatment. There are three plausible explanations for these results. First, carryover effects manifested in unmeasured life history traits such as adult survivorship and life span; in this case, carryover effects are unlikely to manifest in sex-specific trait such as reproductive investment (e.g.–egg production) because there was no difference in the effect of high temperature exposure between males and females. A second possible explanation is compensatory feeding. Individuals were provided with food at replete levels both during and after exposure to high temperatures. However, while compensatory feeding during the copepodite stages may have reduced effects of exposure to the warming treatment on body size and development time, it would likely not have masked potential effects on respiration rate. Any treatment-induced changes in energetic demand may have been compensated for by increased feeding rates in the naupliar stages before molting to C1, however, when respiration measurements were started. *Acartia* species are known to exhibit compensatory feeding in response to reduced nutritional value of prey [41] and salinity shock [42]. Furthermore, the effects of environmental stressors such as combined warming and acidification are weaker at higher food levels [43]. The third explanation is that *A. tonsa* nauplii may have high constitutive expression of heat shock proteins to

accommodate the large changes associated with rapid growth, molting, and metamorphosis. Constitutively high expression levels of heat shock proteins during the naupliar stage would not only prime individuals to respond to an acute shock, but would also reduce the relative energetic costs associated with mounting the heat shock response in warming exposed nauplii relative to the controls. Strong evidence shows upregulation of heat shock proteins in *A. tonsa* adults and eggs [27,44], but we are not aware of any work that has explored this capacity in nauplii.

The comparison between warming treatment and control nauplii was the primary objective of this project. The second novel contribution of this work, however, is documenting the onto-genetic patterns in respiration, with measurements made on the same individuals at multiple stages through development. This contrasts with the bulk respiration measurements on different individuals at different developmental stages that have been used previously to describe these patterns [31,35]. By tracking respiration across development, we are able to account for potential differences in developmental patterns between individuals. The usefulness of this approach is apparent when comparing the mass (mg C) vs. mass-specific respiration (mL O2/ hour/mg C) of individuals across development. The average slope of this relationship was -0.43 ±0.06 (mean ± SE, n = 38) but, between individuals, this slope ranged from -1.17 to 0.56. The wide range of observed slopes reinforces how individual-scale respiration measurements are necessary to account for these differences in response to warming exposure. This is especially important when warming events occur on a within-generational time scale, as population responses will be shaped by the magnitude of individual variation in temperature sensitivity, and highlights that broad patterns may not be predictive of populations dynamics over shorter timescales [23].

The ¾ power law has long been applied to describe the relationship between per-capita respiration and body mass of animals (R = $aM^{3/4}$; [28]). While this is typically examined in the context of across-taxa differences, a similar pattern plays out during development when respiration increases at a slower rate than mass [45]. From this, the expected exponential value for the mass-specific case is -0.25 [28]. We found an exponential value of -0.45 ($r^2$ = 0.32, p<0.0001), which suggests that weight-specific respiration rate decreased faster with mass in our study than generally observed (Fig 5). Despite the considerable uncertainty in the explained variance in this relationship, a comparison of ontogenetic patterns in respiration will still be useful to identify how the methodology used here compares to previous work. Yet, comparison among studies is often difficult as respiration rate units vary in both scale (per minute, per hour, per day) and unit (O2/individual, /unit dry weight, /unit carbon). Additionally, studies in which length is used to estimate weight of individuals often use different conversion factors, making direct comparisons impossible [31,36]. Hence, here we emphasize the comparison of patterns through ontogeny while understanding that specific respiration rate measurements are highly variable to the specific parameters and organisms tested.

The first component of ontogenetic patterns in respiration to compare is the increase in mass-specific respiration from naupliar to copepodite stages. Previous work has demonstrated a large metabolic increase from the naupliar to copepodite stages [31]. We also observed an increase in mass-specific respiration between naupliar and C1 copepodite stages, which is consistent with the observations of Epp & Lewis (1980) on a tropical freshwater calanoid (*Noto-diaptomus venezolanus*). Because of the differences in species and environments, respiration rate comparisons between these studies seem unwarranted. A direct rate comparison of per capita respiration is possible, however, with a study on *A. tonsa* respiration rates in response to starvation [36]. This study used a flow-through respirometry technique with 5 adult females per measurement, and found non-starved adult females (C6) respired 0.95±0.17 nL O2 individual$^{-1}$ min$^{-1}$ (mean±S.E.). This is much smaller than our observed value of 7.75±0.44 nL O2

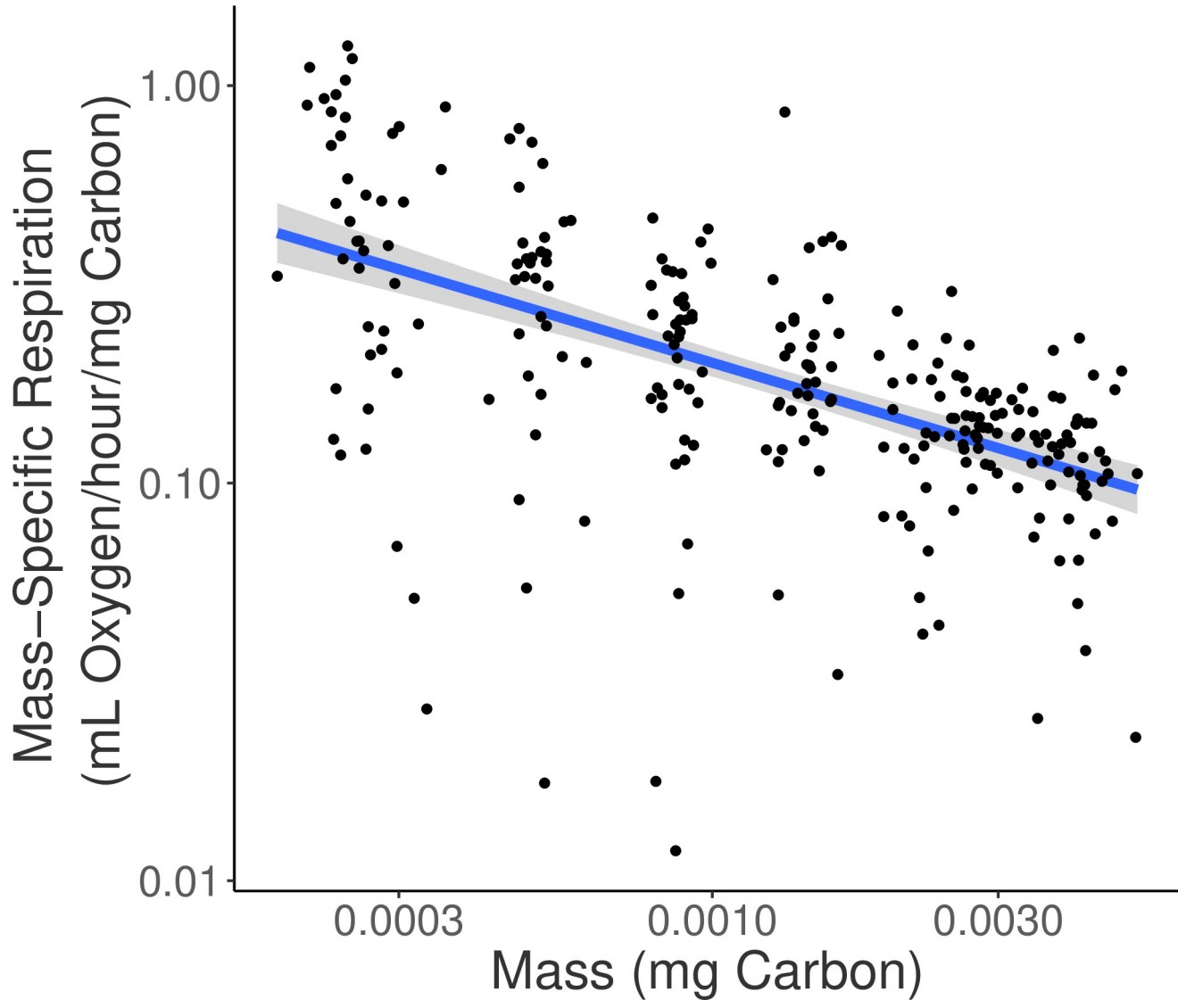

**Fig 5. Relationship between mass-specific respiration (mL Oxygen/hour/mg Carbon) and copepodite mass (mg Carbon).** Scatter plot of $\log_{10}$ mass-specific respiration (mL $O_2$/hour/ng carbon) against mass (mg carbon). Log regression line is described by $y = x^{-0.45}$, $r^2 = 0.31$, $p < 2.2 \times 10^{-16}$, n = 262.

individual$^{-1}$ min$^{-1}$ (mean±S.E.) for adult females at 18°C. Although the copepods being compared here are both *A. tonsa*, this comparison does not account for differences in body mass (as the rates compared are per capita and not mass-specific respiration). Furthermore, past works show that *A. tonsa* from the Baltic region (as used in the previous study; [36]) are generally deeply genetically diverged from *A. tonsa* lineages commonly found in the Western Atlantic and thus there may be differences in baseline metabolic rates between population which this comparison cannot account for [46,47]. While direct comparisons remain difficult due to a variety of factors that cannot be fully accounted for, the patterns we see are overall consistent with previous works.

Several factors can make direct comparison of respiration rates from different studies difficult. We therefore assessed the validity of our mass-specific respiration rates by exploring if respiration rate can meet feeding demands. We focus on adult female rates as they are the

most commonly measured group in other studies [27,31,36]. Our experimentally derived mass-specific respiration value of 0.12 mL O2/mg C/hour equates to 2.88 mL O2/mg C/day. Oxygen respiration can then be converted into respired carbon using a respiratory quotient (RQ) between 0.8 and 1.0. i.e., respired carbon = respired oxygen x RQ x 12/22.4 [48]. Thus, in our study females respired 1.23 to 1.54 mg C food /mg C copepod /day, which is equivalent to 123%-154% of their body weight in carbon per day. To sustain such costs, the copepods must ingest approximately three times this amount of carbon per day [49], equivalent to 369% 462% of their body weight in carbon. Given the average weight of adult females in our study is ~4 μg C (Fig 2), the required amount of carbon ingested per day is 14.76–18.48 μg C, which aligns with prior work that measured ingestion rates of adult female *A. tonsa* at replete food levels between ~10–20 μg C/female/day [50].

This study represents a step toward understanding the full effects of acute naupliar warming exposure on copepod ontogeny. If this experimental design is to be applied to the real-world environment, we recommend additions to the methodology. Primarily, increasing the time of exposure from 1 day to 5 or more days, to match the accepted definition of marine heat waves, which would increase the applicability of this design to real-world warming scenarios. Additionally, incorporating measurements of more life-history traits (e.g., fecundity and egg hatching success, grazing rate), which would provide a more comprehensive view of the organismal response to acute warming exposure through ontogeny. And finally, replicating this experimental design across food concentrations or types [51,52] could highlight compensatory feeding in response to the acute warming exposure as indicated by our conversion of mass-specific respiration rates into ingestion rates.

In summary, our results show that the ecologically important coastal copepod *Acartia tonsa* is metabolically resilient to acute naupliar warming. We observed no carryover effects throughout development of a metabolically impactful naupliar temperature exposure, with no differences in respiration and body size through development or copepodite development time. This resilience may be important for fisheries recruitment and the global biological carbon cycle in a changing climate, as these are powered at least in part by copepod-mediated processes.

## Supporting information

**S1 File. Contains all supporting figures and tables.**
(DOCX)

## Acknowledgments

We thank Catherine Matassa and George McManus for their useful feedback.

## Author Contributions

**Conceptualization:** Mathew Holmes-Hackerd, Matthew Sasaki, Hans G. Dam.

**Formal analysis:** Mathew Holmes-Hackerd, Matthew Sasaki.

**Funding acquisition:** Matthew Sasaki, Hans G. Dam.

**Methodology:** Matthew Sasaki.

**Resources:** Hans G. Dam.

**Supervision:** Matthew Sasaki, Hans G. Dam.

**Visualization:** Mathew Holmes-Hackerd, Matthew Sasaki.

**Writing – original draft:** Mathew Holmes-Hackerd.

**Writing – review & editing:** Mathew Holmes-Hackerd, Matthew Sasaki, Hans G. Dam.

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
