## [Decision Letter · Decision Letter 0]

20 Mar 2023

PONE-D-23-04134Naupliar exposure to acute warming does not affect ontogenetic patterns in respiration, body size, or development time in the cosmopolitan copepod Acartia tonsaPLOS ONE

Dear Dr. Sasaki, 

Thank you for submitting your manuscript to PLOS ONE. After careful consideration, we feel that it has merit but does not fully meet PLOS ONE’s publication criteria as it currently stands. Therefore, we invite you to submit a revised version of the manuscript that addresses the points raised during the review process.

We look forward to receiving your revised manuscript.

Kind regards,

Ram Kumar, Ph.D.

Academic Editor

PLOS ONE

Journal Requirements:

   "We thank Catherine Matassa and George McManus for their useful feedback. This research was supported by a National Science Foundation Grant (OCE 1947965) awarded to H. G. Dam."

   "This research was supported by a National Science Foundation Grant (OCE 1947965) awarded to HGD. The funders had no role in study design, data collection and analysis, decision to publish, or preparation of the manuscript"

3. Please ensure that you include a title page within your main document. You should list all authors and all affiliations as per our author instructions and clearly indicate the corresponding author.

Additional Editor Comments:

The manuscript gives new insight into the impact of rising temperature under climate change on the naupliar development. I would specially like to mention that, this research has adopted an experimental approach to address the effects of temperature rise on the energetics, metabolism and different larval stages. Though the experimental findings are difficult to acclimatize into real world environment, a setup of multiple test combinations have answered many questions and based on this observation the authors were able to demonstrate that, the ecologically important coastal copepod Acartia tonsa exhibits resilience to acute warming. Temperature effects on developmental rates has been elucidated in several papers on P. annandalei , Acartia bilobata etc. The Ms provides additonal information on impacts of acute level temprature change on crustacean larval development

The manuscript will be benefitted from following reference on Nauplius size growth rate: Journal of Plankton Research, Volume 20, Issue 2, 1998, Pages 271–287, https://doi.org/10.1093/plankt/20.2.271 and

DOI: 10.1127/0003-9136/2003/0157-0351

Authors should carefully check formatting patterns and reference style

Reviewers' comments:

Reviewer's Responses to Questions

**Comments to the Author**

1. Is the manuscript technically sound, and do the data support the conclusions?

Reviewer #1: Yes

Reviewer #2: Yes

2. Has the statistical analysis been performed appropriately and rigorously? 

Reviewer #1: Yes

Reviewer #2: Yes

3. Have the authors made all data underlying the findings in their manuscript fully available?

Reviewer #1: Yes

Reviewer #2: Yes

4. Is the manuscript presented in an intelligible fashion and written in standard English?

Reviewer #1: Yes

Reviewer #2: Yes

5. Review Comments to the Author

Reviewer #1: Dear Authors,

I found this manuscript very interesting. However, there are some comments and suggestions to clarify some part to improve the manuscript.

1. Line no: 107-109 should go to conclusion section of manuscript.

2. All the figure and Graphs needs to be improved. It is not clear why author used scale such as 0e+00, 3e04….. in figure 3

and 5. Also, in Figure 1, in x-axis labels, the unit of temperature is written as (C) and (degree C). It should be (°C).

3. As this study has also analysed the sex-dependent metabolic responses of copepod but, it is not clear how many female

or male individuals were present in the experiment or replicates taken.

4. In Copepodite respiration rate, in material and methods section, line no:183 is not clear.

(a) The author should specify if they measured the respiration of copepodite stage in 8-hour interval or was it measured for

a time period of 8 hours.

(b) If it measured for a time period of 8 hours, a description should be added about why only 8 hours of respiration

analyses were done for copepodite.

5. Line no: 420 In equation in place of ‘*’ should use ‘x’ (character code: 00D7).

6. Line 381-383 and line 413-415 seems like repetition.

7. Formatting of references need to be uniform and according to journal reference style.

Reviewer #2: Manuscripts Number: PONE-D-22-29055

Reviewer’s comment

The manuscript is well written and provides new insight on the impact of rising temperature under climate change on the larval behavior of copepod. This experimental work gives us new understanding about the potential of copepods larvae resilience. Knowing the importance of the organism in the trophic structure and energy transfer and the information on how climate induced changes may lead to a cascading effect (top-down or bottom-up) is the need of hour. I would specially like to mention that, this research has adopted an experimental approach to address the effects of temperature rise on the energetics, metabolism and different larval stages. Though the experimental findings are difficult to acclimatize into real world environment, a setup of multiple test combinations have answered many questions and based on this observation the authors were able to demonstrate that, the ecologically important coastal copepod Acartia tonsa exhibits resilience to acute warming.

I believe under ever-changing climate and ever-increasing human pressure on the world global ecosystem, this manuscript will fall in the line of global efforts of ecosystem conservation and future prediction model. This work also provided a promising finding which is quite opposite to the prevailing notion on the adverse impact due to the climate change which must be supported and substantiated later. Therefore, I recommend the manuscript has a potential for publication in PLOSE ONE. However, few of minor corrections from my side have been highlighted, corrected or deleted in the PDF file and request the authors to kindly incorporate the same as long as those are empirical. Since I am not a native of English speaking but still, I gave a try to correct few.

However, I want a clarification from the author’s side

“Since the authors have used sub-lethal dose of temperature and such mild level of treatment may not be effective and proven fatal at all stages? My question is what was the rationale for setting the particular level of temperature?

6. PLOS authors have the option to publish the peer review history of their article (what does this mean?). If published, this will include your full peer review and any attached files.

Reviewer #1: **Yes: **Devesh Kumar Yadav

Reviewer #2: **Yes: **Jawed Equbal

---

## [Author Response · Author response to Decision Letter 0]

26 Mar 2023

We have included responses to the reviewer comments in a separate file. If needed, we have copied the contents of that file below. 

Response to editor and reviewer comments

Editor Comments:

The manuscript gives new insight into the impact of rising temperature under climate change on the naupliar development. I would specially like to mention that, this research has adopted an experimental approach to address the effects of temperature rise on the energetics, metabolism and different larval stages. Though the experimental findings are difficult to acclimatize into real world environment, a setup of multiple test combinations have answered many questions and based on this observation the authors were able to demonstrate that, the ecologically important coastal copepod Acartia tonsa exhibits resilience to acute warming. Temperature effects on developmental rates has been elucidated in several papers on P. annandalei , Acartia bilobata etc. The Ms provides additonal information on impacts of acute level temprature change on crustacean larval development. The manuscript will be benefitted from following reference on Nauplius size growth rate: Journal of Plankton Research, Volume 20, Issue 2, 1998, Pages 271–287, https://doi.org/10.1093/plankt/20.2.271 and

DOI: 10.1127/0003-9136/2003/0157-0351

Authors should carefully check formatting patterns and reference style.

We thank the editor for their support of our manuscript. We have addressed the two formatting issues in our previous submission by including a title page, correcting the section heading formats, and including captions for the Supporting Information at the end of the manuscript text. We have also corrected the reference formatting. 

Additionally, we included the references to the specified studies in the discussion section of our manuscript (line 442). We thank the editor for bringing these studies to our attention as they illustrate an important additional consideration about developmental trajectories in copepods. 

 

Reviewer #1: 

Dear Authors,

I found this manuscript very interesting. However, there are some comments and suggestions to clarify some part to improve the manuscript.

We thank the reviewer for their helpful feedback on our manuscript. We have incorporated the suggestions into the revised version as detailed below. 

1. Line no: 107-109 should go to conclusion section of manuscript.

We have moved this sentence to the discussion section of the manuscript (now found at line 331-332). 

2. All the figure and Graphs needs to be improved. It is not clear why author used scale such as 0e+00, 3e04….. in figure 3 and 5. Also, in Figure 1, in x-axis labels, the unit of temperature is written as (C) and (degree C). It should be (°C).

We have modified the y-axis scale to use decimal values instead of scientific notation. We have also corrected the x-axis labels in Figure 1. 

3. As this study has also analysed the sex-dependent metabolic responses of copepod but, it is not clear how many female or male individuals were present in the experiment or replicates taken.

We have now included the number of male and female individuals in the results section (lines 243-245). 

4. In Copepodite respiration rate, in material and methods section, line no:183 is not clear.

(a) The author should specify if they measured the respiration of copepodite stage in 8-hour interval or was it measured for a time period of 8 hours.

(b) If it measured for a time period of 8 hours, a description should be added about why only 8 hours of respiration analyses were done for copepodite.

We have clarified the language in this section to specify that respiration rates were measured over an 8 hour period of time, not at 8 hour intervals (line 185-188). We used this duration because it minimized the amount of experimental handling each individual was subjected to, while still resulting in detectable oxygen drawdown. 

5. Line no: 420 In equation in place of ‘*’ should use ‘x’ (character code: 00D7).

We have replaced the character as suggested (line 426).

6. Line 381-383 and line 413-415 seems like repetition.

We have re-written the second section (now found at line 420) to avoid repetition. 

7. Formatting of references need to be uniform and according to journal reference style.

We have corrected the reference formatted in this new version. 

 

Reviewer #2: 

The manuscript is well written and provides new insight on the impact of rising temperature under climate change on the larval behavior of copepod. This experimental work gives us new understanding about the potential of copepods larvae resilience. Knowing the importance of the organism in the trophic structure and energy transfer and the information on how climate induced changes may lead to a cascading effect (top-down or bottom-up) is the need of hour. I would specially like to mention that, this research has adopted an experimental approach to address the effects of temperature rise on the energetics, metabolism and different larval stages. Though the experimental findings are difficult to acclimatize into real world environment, a setup of multiple test combinations have answered many questions and based on this observation the authors were able to demonstrate that, the ecologically important coastal copepod Acartia tonsa exhibits resilience to acute warming.

I believe under ever-changing climate and ever-increasing human pressure on the world global ecosystem, this manuscript will fall in the line of global efforts of ecosystem conservation and future prediction model. This work also provided a promising finding which is quite opposite to the prevailing notion on the adverse impact due to the climate change which must be supported and substantiated later. Therefore, I recommend the manuscript has a potential for publication in PLOSE ONE. However, few of minor corrections from my side have been highlighted, corrected or deleted in the PDF file and request the authors to kindly incorporate the same as long as those are empirical. Since I am not a native of English speaking but still, I gave a try to correct few.

However, I want a clarification from the author’s side: “Since the authors have used sub-lethal dose of temperature and such mild level of treatment may not be effective and proven fatal at all stages? My question is what was the rationale for setting the particular level of temperature?

We thank the reviewer for their helpful comments and suggestions on the manuscript – we have incorporated many of the recommended changes into the resubmitted version (see lines 18-20, 20-22, 28-29, 35-37 for examples). We have also expanded the sections detailing our decision to focus on sub-lethal stress in the methods section (lines 130-132). Namely, selection of a non-lethal temperature was necessary in order to avoid confounding effects of selection for heat tolerant individuals on measurements later in development.

---

## [Decision Letter · Decision Letter 1]

10 Apr 2023

Naupliar exposure to acute warming does not affect ontogenetic patterns in respiration, body size, or development time in the cosmopolitan copepod *Acartia tonsa*

PONE-D-23-04134R1

Dear Dr. Sasaki,

We’re pleased to inform you that your manuscript has been judged scientifically suitable for publication and will be formally accepted for publication once it meets all outstanding technical requirements. We thank you for considering Plose-One as an important vehicle for dissemination of your research findings.  

Kind regards,

Ram Kumar, Ph.D.

Academic Editor

PLOS ONE

---

## [Editor Report · Acceptance letter]

12 Apr 2023

PONE-D-23-04134R1 

Naupliar exposure to acute warming does not affect ontogenetic patterns in respiration, body size, or development time in the cosmopolitan copepod *Acartia tonsa*

Dear Dr. Sasaki:

I'm pleased to inform you that your manuscript has been deemed suitable for publication in PLOS ONE. Congratulations! Your manuscript is now with our production department. 

Kind regards, 

on behalf of

Professor Ram Kumar 

Academic Editor

PLOS ONE